# Kernel similarity matching with Hebbian Networks

**Kyle Luther**
Princeton Neuroscience Institute
kluther@princeton.edu

**H. Sebastian Seung**
Princeton Neuroscience Institute
Princeton Computer Science Department
sseung@princeton.edu

## Abstract

Recent works have derived neural networks with online correlation-based learning rules to perform *kernel similarity matching*. These works applied existing linear similarity matching algorithms to nonlinear features generated with random Fourier methods. In this paper we attempt to perform kernel similarity matching by directly learning the nonlinear features. Our algorithm proceeds by deriving and then minimizing an upper bound for the sum of squared errors between output and input kernel similarities. The construction of our upper bound leads to online correlation-based learning rules which can be implemented with a 1 layer recurrent neural network. In addition to generating high-dimensional linearly separable representations, we show that our upper bound naturally yields representations which are sparse and selective for specific input patterns. We compare the approximation quality of our method to neural random Fourier method and variants of the popular but non-biological "Nyström" method for approximating the kernel matrix. Our method appears to be comparable or better than randomly sampled Nyström methods when the outputs are relatively low dimensional (although still potentially higher dimensional than the inputs) but less faithful when the outputs are very high dimensional.

## 1   Introduction

Brain inspired learning algorithms have a long history in the field of neural networks and machine learning [Rosenblatt, 1958, Olshausen and Field, 1996, Lee and Seung, Riesenhuber and Poggio, 1999, Hinton, 2007, Lillicrap et al., 2016]. While many algorithms have diverged from their biological roots, the motivation to study biology remains clear: the human brain is such a powerful learning agent, there must be insights to be gained by making our algorithms look "brain-like". This paper is focused on merging biological constraints with the well-established field of kernel-based machine learning.

A common assumption in brain-inspired models of learning is that synaptic update rules should be a) online, meaning the algorithm only has access to a single input pattern at a time and b) local, meaning synapses should only be modified using information immediately available to the synapse, often just the pre- and post-firing rates of the neurons to which it is connected. Learning rules with these properties are commonly referred to as Hebbian learning rules [Chklovskii, 2016].

Recent works have devised neural networks with Hebbian learning rules that perform linear similarity matching. These networks map every input $\mathbf{x}_t$ to a representation $\mathbf{y}_t$ such that linear output similarities match linear input similarities $\mathbf{y}_s \cdot \mathbf{y}_t \approx \mathbf{x}_s \cdot \mathbf{x}_t$. These networks are interesting as models for real brains because they display a number of interesting biological properties: they are recurrent networks with correlation-based learning rules [Pehlevan et al., 2018] and can be modified to include non-negativity [Pehlevan and Chklovskii, 2014], sparsity, and convolutional structure [Obeid et al., 2019].

However there is a problem if one believes these networks should ultimately generate representations which are useful for downstream tasks. If similarities are actually matched, that is if $\mathbf{y}_s \cdot \mathbf{y}_t = \mathbf{x}_s \cdot \mathbf{x}_t$,

then the outputs are simply an orthogonal transformation of the inputs, $\mathbf{y}_t = \mathbf{Q}\mathbf{x}_t$, which is unlikely to have significant impact on downstream tasks. Bahroun et al. [2017] identified this problem and proposed a solution: instead of matching linear input similarities, one can match nonlinear input similarities: $\mathbf{y}_s \cdot \mathbf{y}_t \approx f(\mathbf{x}_s, \mathbf{x}_t)$. The authors provided a method that can be applied to any shift-invariant kernel. They applied the random Fourier feature method of Rahimi et al. [2007] to map inputs to nonlinear feature vectors $\mathbf{x} \rightarrow \psi$ and then applied the linear similarity matching framework of Pehlevan et al. [2018] to these nonlinear features.

In this paper, we tackle the same neural kernel similarity matching problem with a different approach. Instead of using random nonlinear features, we directly optimize for the features with Hebbian learning rules that resemble the learning rules derived in the original works on linear similarity matching. To derive our learning rules, we show that for any kernel we can upper bound the sum of squared errors $|\mathbf{y}_s \cdot \mathbf{y}_t - f(\mathbf{x}_s, \mathbf{x}_t)|^2$ with a correlation-based energy. Gradient-based optimization of our upper bound with will lead to a neural network with correlation-based learning rules.

## 2  Correlation-based bound for kernel similarity matching

**Roadmap for this section** We first define the kernel similarity matching problem (Eq. 1). We then derive a correlation-based optimization (Eq. 6) which is an upper bound for to Eq. 1 (up to a constant that does not depend on the representations). We then use a Legendre transform to derive an equivalent (except for the numerical stability parameter $\lambda$) optimization problem in Eq. 9 that will lend itself towards online updates.

**Kernel similarity matching** Assume we are given a set of input vectors $\{\mathbf{x}^t \in \mathbb{R}^M\}_{t=1}^T$ and a positive semi-definite kernel function $f : \mathbb{R}^M \times \mathbb{R}^M \rightarrow \mathbb{R}$ which defines the similarity between input vectors. The goal is to find a corresponding set of representations $\{\mathbf{y}^t \in \mathbb{R}^N\}_{t=1}^T$ such that for all pairs $(s, t)$ we have $\mathbf{y}^s \cdot \mathbf{y}^t \approx f(\mathbf{x}^s, \mathbf{x}^t)$. We will assume that $T \gg N > M$: the dimensionalities of $\mathbf{x}, \mathbf{y}$ are much lower than the number of samples $T$, but $\mathbf{y}$ are still higher dimensional than the inputs. This is formalized by minimizing the sum of squared errors:

$$\min_{\{\mathbf{y}^t\}} \frac{1}{T^2} \sum_{s,t}^T \left[ f(\mathbf{x}^s, \mathbf{x}^t) - \mathbf{y}^s \cdot \mathbf{y}^t \right]^2 \tag{1}$$

This is known as the classical multidimensional scaling objective [Borg and Groenen, 2005]. For arbitrary nonlinearity $f$ this can be solved exactly by finding the top $N$ eigenvectors of the $T \times T$ input similarity matrix [Borg and Groenen, 2005], and is therefore closely related to kernel PCA [Schölkopf et al., 1997]. However, this requires computing and storing similarities for all pairs of input vectors which breaks the online constraint that we require for biological realism. The purpose of this paper is to find an online algorithm, with correlation-based computations, that can at least approximately minimize Eq. 1.

**Correlation based upper bound** In this section we provide an upper bound to Eq. 1 which does not require computing $f(\mathbf{x}_s, \mathbf{x}_t)$ for any $(s, t)$. The first step is to expand the square in Eq (1) to yield:

$$\frac{1}{T^2} \sum_{s,t}^T \left[ f(\mathbf{x}^s, \mathbf{x}^t) - \mathbf{y}^s \cdot \mathbf{y}^t \right]^2 = -\frac{2}{T^2} \sum_{s,t} f(\mathbf{x}^s, \mathbf{x}^t) \mathbf{y}^s \cdot \mathbf{y}^t + \frac{1}{T^2} \sum_{s,t} (\mathbf{y}^s \cdot \mathbf{y}^t)^2 + \text{const} \tag{2}$$

We will now show how to bound the first term on the right hand side.

**Theorem 1.** *If $f$ is a positive semidefinite kernel function, then*

$$\frac{1}{2T^2} \sum_{s,t} y^s y^t f(\mathbf{x}^s, \mathbf{x}^t) \geq \frac{1}{T} \sum_t q y^t f(\mathbf{x}^t, \mathbf{w}) - \frac{1}{2} q^2 f(\mathbf{w}, \mathbf{w}) \tag{3}$$

*for all $q$ and $\mathbf{w}$.*

*Proof.* Because $f$ is a positive semi-definite kernel, we can assign to any set of $M$-dimensional vectors $\{\mathbf{w}, \mathbf{x}_1, \dots, \mathbf{x}_T\}$, a corresponding set of (at most) $T + 1$-dimensional vectors $\{\phi_{\mathbf{w}}, \phi_1, \dots, \phi_T\}$ whose inner products yield the similarity defined by $f$:

$$\phi_t \cdot \phi_{t'} = f(\mathbf{x}_t, \mathbf{x}_{t'}) \qquad \phi_t \cdot \phi_{\mathbf{w}} = f(\mathbf{x}_t, \mathbf{w}) \qquad \phi_{\mathbf{w}} \cdot \phi_{\mathbf{w}} = f(\mathbf{w}, \mathbf{w}) \tag{4}$$

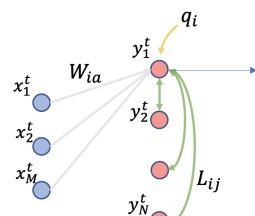

**(a) network architecture**

**(b) recurrent network dynamics**

$$\dot{y}_i^t = q_i f(\mathbf{w}_i, \mathbf{x}^t) - \sum_j L_{ij} y_j^t - \lambda y_i^t$$

**(c) steady state network output**

$$y_i^t = \sum_j f(\mathbf{w}_i, \mathbf{x}^t) q_j [(\mathbf{L} + \lambda \mathbf{I})^{-1}]_{ij}$$

**(d) Hebbian update rules for Gaussian kernel**

$$f(\mathbf{u}, \mathbf{v}) = e^{-\gamma \|\mathbf{u}-\mathbf{v}\|^2}$$

$$\dot{W}_{ia} \propto [y_i f_i'] x_a - [y_i f_i'] W_{ia}$$
$$\dot{q}_i \propto y_i f_i - q_i$$
$$\dot{L}_{ij} \propto y_i y_j - L_{ij}$$

Figure 1: Neural network implementation of the optimization in Eq. 9 (a) network architecture (b) recurrent network dynamics (c) steady state network response (d) Hebbian update rules for the special case of Gaussian kernels (the precise form of these updates will be depend on the kernel)

Now consider the vector difference $\frac{1}{T} \sum_t y_t \boldsymbol{\phi}_t - q\boldsymbol{\phi}_w$. The squared norm of this difference is of course non-negative. Additionally we can expand out this square:

$$0 \leq \frac{1}{2} \left\| \frac{1}{T} \sum_t y_t \boldsymbol{\phi}_t - q\boldsymbol{\phi}_\mathbf{w} \right\|^2 = \frac{1}{2T^2} \sum_{s,t} y_s y_t \boldsymbol{\phi}_s \cdot \boldsymbol{\phi}_t - \frac{1}{T} \sum_t q y_t \boldsymbol{\phi}_t \cdot \boldsymbol{\phi}_\mathbf{w} + \frac{1}{2} q^2 \boldsymbol{\phi}_\mathbf{w} \cdot \boldsymbol{\phi}_\mathbf{w} \quad (5)$$

At this point we can simply replace all dot products with the equivalent nonlinear similarities $f(\cdot, \cdot)$ in Equation 4 and rearrange the terms to yield our key inequality (Eq. 3). $\qquad\square$

Our inequality (Eq. 3) still holds if we maximize the right hand side with respect to $q$ and $\mathbf{w}$. For every index $i$ of $\mathbf{y}$, we find the optimal $\mathbf{w}_i, q_i$, and then replace the first pairwise sum in Eq. (2) with our upper bounds. Additionally we rearrange the order of the summations in second term on the right hand side of Eq. (2) to yield the following upper bound for the $y$-dependent terms in Eq. (2):

$$\min_{\mathbf{y}^t} \min_{q_i, \mathbf{w}_i} -\frac{1}{T} \sum_{t=1}^{T} \sum_{i=1}^{N} \left[ q_i y_i^t f(\mathbf{w}_i, \mathbf{x}^t) - \frac{1}{2} q_i^2 f(\mathbf{w}_i, \mathbf{w}_i) \right] + \frac{1}{4} \sum_{i,j=1}^{N} \left( \frac{1}{T} \sum_{t=1}^{T} y_i^t y_j^t \right)^2 \quad (6)$$

**Online focused reformulation** We can further remove the square of the correlation matrix $\frac{1}{T} \sum_t y_i^t y_j^t$ (another impediment to online learning) by introducing a Legendre transformation: $\frac{1}{2} C_{ij}^2 \rightarrow \max_{L_{ij}} C_{ij} L_{ij} - \frac{1}{2} L_{ij}^2$:

$$\min_{\mathbf{W}, \mathbf{Y}, \mathbf{q}} \max_{\mathbf{L}} \frac{1}{T} \sum_{t=1}^{T} \left[ -\sum_{i=1}^{N} \left[ q_i y_i^t f(\mathbf{w}_i, \mathbf{x}^t) - \frac{1}{2} q_i^2 f(\mathbf{w}_i, \mathbf{w}_i) \right] + \frac{1}{2} \sum_{i,j=1}^{N} \left[ L_{ij} y_i^t y_j^t - \frac{1}{2} L_{ij}^2 \right] \right] \quad (7)$$

We can swap the order of the $y$ and $L$ optimizations, because the objective obeys the strong min-max property with $\mathbf{W}, \mathbf{q}$ fixed (Appendix Section A of Pehlevan et al. [2018]). We add one final term $\frac{\lambda}{NT} \sum_{t=1}^{T} \sum_{i=1}^{N} (y_i^t)^2$ to the objective, which can be important for numerical stability of our resulting algorithm. In our experiments $\lambda = 0.001$. Finally, to better motivate our online algorithm, we define the "per-sample-energy":

$$e^t := \sum_{i=1}^{N} -\left[ q_i y_i^t f(\mathbf{w}_i, \mathbf{x}^t) - \frac{1}{2} q_i^2 f(\mathbf{w}_i, \mathbf{w}_i) \right] + \frac{1}{2} \sum_{i,j=1}^{N} \left[ L_{ij} y_i^t y_j^t - \frac{1}{2} L_{ij}^2 \right] + \frac{\lambda}{2} \sum_{i=1}^{N} (y_i^t)^2 \quad (8)$$

where $e^t := e(\mathbf{y}^t, \mathbf{x}^t; \mathbf{W}, \mathbf{q}, \mathbf{L})$. The final optimization we will perform, which is equivalent to the optimization in Eq. 6, and is derived as an upper bound to Eq. 1, is thus:

$$\min_{\mathbf{W}, \mathbf{q}} \max_{\mathbf{L}} \min_{\mathbf{Y}} \frac{1}{T} \sum_{t=1}^{T} e(\mathbf{y}^t, \mathbf{x}^t; \mathbf{W}, \mathbf{q}, \mathbf{L}) \quad (9)$$

## 3   Neural network optimization

Applying a stochastic gradient descent-ascent algorithm to Eq. (9) yields a neural network (Fig. 1) in which $y_i^t$ is the response of neuron $i$ to input pattern $t$, $\mathbf{w}_i$ is the vector of incoming connections to

neuron $i$ from the input layer, $q_i$ is a term which modulates the strength of these incoming connections, and $L_{ij}$ is a matrix of lateral recurrent connections between outputs.

Specifically for the neural algorithm we initialize $W_{ia} \leftarrow \mathcal{N}(0, 1)$, $q_i \leftarrow 1$ and $L_{ij} \leftarrow \mathbf{I}_{ij}$. At each iteration sample a minibatch of inputs $\{\mathbf{x}^b\}$. Using Eq.12 we compute the $\{\mathbf{y}^b\}$ which minimize Eq. 9 for fixed synapses. Using these optimal $\{\mathbf{y}^b\}$, we compute the minibatch energy $e = \frac{1}{B}\sum_b e(\mathbf{x}^b, \mathbf{y}^b; \mathbf{W}, \mathbf{q}, \mathbf{L})$ and take a gradient descent step for $\mathbf{q}$, a rescaled gradient descent step for $\mathbf{w}$ and a gradient ascent step for $\mathbf{L}$:

$$\mathbf{w}_i \leftarrow \mathbf{w}_i - \frac{\eta_w}{q_i^2}\frac{\partial e}{\partial \mathbf{w}_i} \qquad q_i \leftarrow q_i - \eta_q \frac{\partial e}{\partial q_i} \qquad L_{ij} \leftarrow L_{ij} + \eta_l \frac{\partial e}{\partial L_{ij}} \qquad (10)$$

**Convergence of the neural algorithm** We treat convergence of this gradient descent ascent algorithm as an empirical issue. We adopt the "two time scale" strategy that has shown empirical successes for training generative adversarial networks [Heusel et al., 2017]. We choose the learning rates such that $\eta_q, \eta_w \ll \eta_l$. Intuitively when choosing $\eta_l$ to be large, the $L_{ij}$ can approximately maximize Eq. 9 for any particular $\mathbf{q}, \mathbf{W}$ so that the min-max ordering is roughly preserved. In practice this ratio is important for convergence. We do not observe convergence when the ratios $\eta_w/\eta_l$ or $\eta_q/\eta_l$ are large. Unfortunately it is an empirical question of what is "too large". If we could show that the objective were concave in $L$, it can be gradient descent ascent with smaller learning rates for $W, q$ would indeed converge to a saddle point [Lin et al., 2020, Seung, 2019]. However, this question will have to be left for future work.

Empirically it is sometimes observed that $q_i$ quickly shrinks to a small value early in training, which subsequently leads to small gradients for $\mathbf{w}$. The rescaling of the $\mathbf{w}_i$ updates provides an adaptive learning rate that appeared to improve training times in practice. We have attached the main portion of the training code, written using PyTorch, in the appendix.

## 3.1 Network dynamics

Assuming fixed parameters $\mathbf{q}, \mathbf{W}, \mathbf{L}$, the gradient for $\mathbf{y}$ can be computed for any input pattern $\mathbf{x}$. Gradient descent can be used to perform the inner loop minimization in Equation 9:

$$\dot{y}_i = \eta_y \left[ q_i f(\mathbf{w}_i, \mathbf{x}) - \sum_{j=1}^N L_{ij} y_j - \lambda y_i \right] \qquad (11)$$

Like previous works on linear similarity matching, these dynamics can be interpreted as the dynamics of a one-layer recurrent neural network with all-to-all inhibition $\sum_{j=1}^N L_{ij} y_j$ between units. A diagram of this network is shown in Figure 1. Note that we can analytically perform the inner loop minimization with a non-neural algorithm:

$$y_i \leftarrow \sum_j [\mathbf{L} + \lambda \mathbf{I}]_{ij}^{-1} q_j f(\mathbf{w}_j, \mathbf{x}) \qquad (12)$$

This is useful both conceptually and for speeding up the training process in our experiments. This formula shows us that $\mathbf{y}$ is a linear function of the non-linear feedforward input $f(\mathbf{w}_i, \mathbf{x}_t)$. This is different from Seung and Zung [2017], Pehlevan and Chklovskii [2014] where the neurons are non-linear functions (due to non-negativity constraints) of linear feedforward input $\mathbf{w}_i \cdot \mathbf{x}_t$.

## 3.2 Synaptic learning rules: arbitrary kernel

In the previous section we saw how the $\mathbf{W}$ could be interpreted as feedforward synapses, $\mathbf{q}$ as feedforward regulatory terms, $\mathbf{L}$ as inhibitory synapses. Gradient descent on $\mathbf{W}, \mathbf{q}$ and gradient ascent on $\mathbf{L}$ provides an algorithm for performing the optimization in Equation 9. At each step, we compute the optimal $\mathbf{y}$. For simplicity, we consider the case with a single input, in which case we drop the index $b$ on $\mathbf{x}^b, \mathbf{y}^b$. The stochastic gradients for $\mathbf{W}$ lead to the update:

$$\Delta \mathbf{w}_i \propto y_i \partial f(\mathbf{w}_i, \mathbf{x})/\partial \mathbf{w}_i - q_i \partial f(\mathbf{w}_i, \mathbf{w}_i)/\partial \mathbf{w}_i \qquad (13)$$

Classically Hebbian rules have been defined so that the update is linear in the input $\mathbf{x}$ (although they can be nonlinear in the output $\mathbf{y}$ which is a function of $\mathbf{x}$) (Eq. 1 of Brito and Gerstner [2016]). This

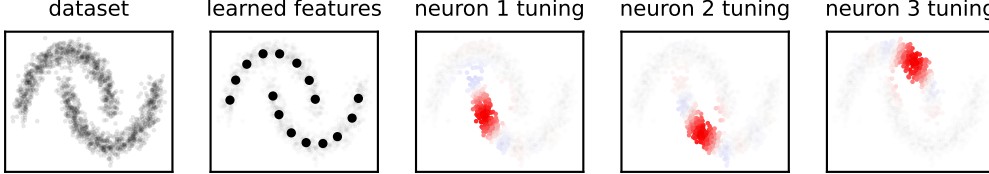

| dataset | learned features | neuron 1 tuning | neuron 2 tuning | neuron 3 tuning |

Figure 2: Overview of our Hebbian radial basis function network on the half moons dataset (a) dataset, (b) features $\{\mathbf{w}_i\}_{i=1}^{16}$ (c,d,e) response profiles of 3 neurons.

rule is more general as it is a nonlinear function of the input vector $\partial f(\mathbf{w}_i, \mathbf{x})/\partial \mathbf{w}_i = h(\mathbf{x}, \mathbf{w}_i)$. However we note that the spirit of Hebb is still here as this is an online, local, correlation-based learning rule.

The regulatory terms (essentially controlling the magnitude of the strength of feedforward input) can be updated with:

$$\Delta q_i \propto y_i f(\mathbf{w}_i, \mathbf{x}) - f(\mathbf{w}_i, \mathbf{w}_i) q_i \tag{14}$$

Here we have the correlation between the feedforward input and the neurons response. Finally gradients for the inhibitory synapses are:

$$\Delta L_{ij} \propto y_i y_j - L_{ij} \tag{15}$$

This is exactly the same "anti-hebbian" update seen in previous linear similarity matching works Pehlevan et al. [2018]. The inhibition grows in strength as the correlation between neurons grows.

### 3.3 Synaptic learning rules: radial basis function kernel

Before moving on, we'll consider the form of the update rules in Eq. 13,14 when the kernel is a radial basis function, i.e. when the kernel is a function of the Euclidean distance. For simplicity we'll also assume the kernel is normalized so that $f(\mathbf{v}, \mathbf{v}) = 1$:

$$f(\mathbf{u}, \mathbf{v}) := g(\|\mathbf{u} - \mathbf{v}\|) \quad \text{and} \quad g(0) = 1 \tag{16}$$

In this case we get the gradient updates for $\mathbf{w}_i, q_i$:

$$\Delta \mathbf{w}_i \propto [y_i g_i']\mathbf{x} - [y_i g_i']\mathbf{w} \quad \Delta q_i \propto y_i g_i - q_i \tag{17}$$

The update for $\mathbf{w}_i$ is proportional to the input $\mathbf{x}$, but modulated by the output response ($y_i$) and a function of the feedforward input ($g_i'$). The updates for $L_{ij}$ do not depend on the form of the kernel.

## 4 Experiments

We train networks using a Gaussian kernel for the half moons dataset and a "power-cosine" kernel (defined in section 4.2) for the MNIST dataset. We compare the approximation error (Eq. 1) of our method to the approximation error given by a) the optimal eigenvector-based solution (which we label as kernel PCA) b) Nyström approximation with uniformly sampled landmarks c) Nyström approximation using KMeans to generate the landmarks d) Nyström approximation using our generated features ($w_i$) as the landmarks and e) random Fourier feature method (This method is not applicable to the cosine-based kernel we use for the MNIST dataset). The "dimensionality" refers to the number of components for the PCA method, the number of landmarks for the Nyström methods, and the number of Fourier features for the Fourier method. See the appendix for more details regarding each of these 5 methods. Method (e) is the only other explicitly neural method.

### 4.1 Half Moons Dataset

We train our algorithm on a simple half moons dataset (which can be generated with Pedregosa et al. [2011]), shown in Figure 2. It consists of 1600 input vectors $\mathbf{x} = [x_1, x_2]$ drawn from a distribution of two noisy interleaving half circles. We use a Gaussian kernel with $\sigma = 0.3$ to measure input

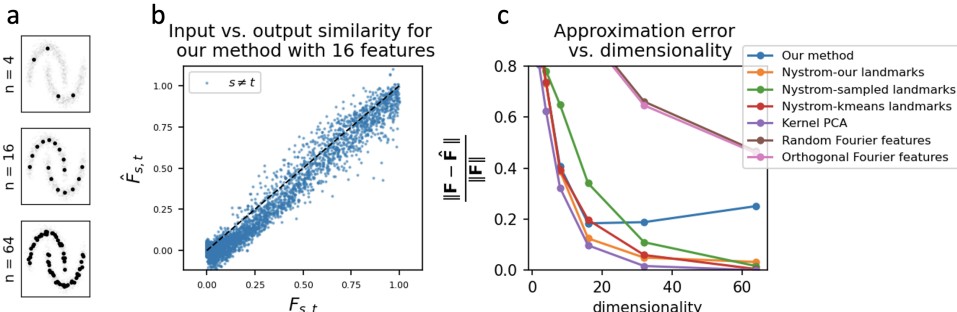

Figure 3: Approximation error vs. dimensionality for the half moons dataset with the Gaussian kernel (a) learned features for $n = 4, 16, 64$ (b) input-output similarities for 16 dimensional networks (c) normalized root-mean-square error between true kernel matrix and various approximation methods. The neural method (dashed blue) that we derive in Section 3 performs well for $n <= 16$, but the approximation actually gets worse as we increase the dimensionality.

similarities: $f(\mathbf{u}, \mathbf{v}) = e^{-\frac{\|\mathbf{u}-\mathbf{v}\|^2}{2\sigma^2}}$. We vary the number of neurons $n \in \{2, 4, 8, 16, 32, 64\}$. See the appendix for training details.

**Emergence of sparse, template-tuned neurons** In Fig. 2 we show the learned features $\{\mathbf{w}_i\}$ when we train our algorithm with 16 neurons. We observe that the features appear to tile the input space. We also show the tuning properties of 3 of the output neurons over the dataset. To generate these figures, we color code each sample in the dataset with the response of neuron $y_i$. Gray indicates zero response, red indicates a positive response and blue indicates a negative response. We observe that neurons appear to respond with large positive values centered around a small localized region of the input dataset. The features closely resemble the cluster centers returned by KMeans.

**Kernel approximation error** In panel (a) of Fig. 3 we show the learned features for $n = \{4, 16, 64\}$. In panel (b) we plot the input similarities vs. output similarities generated by our neural algorithm with 16 dimensional outputs. In panel (c), we plot the normalized mean squared error for our method compared to random Fourier features Rahimi et al. [2007], orthogonal Fourier features Yu et al. [2016], Nyström methods, and kernel PCA. The neural method of Bahroun et al. [2017] gives equivalent results to the random Fourier method. Kernel PCA is optimal, but non-neural.

We observe that for small dimensionality ($n \leq 16$) our method actually seems to marginally outperform the Nyström+KMeans method, which outperforms the Nyström+randomly sampled landmarks method. Additionally, using the Nyström approximation with our features seems to be uniformly better than the representations we generate with the neural net. Essentially, our algorithm learns useful landmarks, but for most faithful representation, it is better to just throw away the neural responses and simply use the Nyström approximation with our landmarks. It is worth mentioning that as you increase the dimensionality higher, the Random Fourier method ultimately does converge to zero error, unlike our method.

**Utility of representations evaluated by KMeans clustering** In Fig. 4 we visualize the principle components of the inputs $\mathbf{x}$ and 16D representations $\mathbf{y}$. Of course, the principle components of $\mathbf{x}$ are not too interesting, they are just a reflected version of the original 2D dataset. The top two components of $\mathbf{y}$ are more linearly separable than the inputs and this indicates that a strong nonlinear transformation has been applied to the inputs. Additionally, we run KMeans on $\mathbf{x}$ and on $\mathbf{y}$ (we use the implementation of scikit-learn, and take the lowest energy solution using 100 inits). We observe that the clustering yields the nearly perfect labels when performed on $\mathbf{y}$. The kernel similarity matching vectors appear to be better suited for downstream learning tasks than the original inputs.

## 4.2 MNIST Dataset

We train our algorithm on the MNIST handwritten digits dataset LeCun [1998]. The dataset consists of 70,000 images of 28x28 handwritten digits, which we cropped by 4 pixels on each side to yield 20x20 images (which become 400 dimensional inputs). We use kernels of the form $f(\mathbf{u}, \mathbf{v}) = \|\mathbf{u}\|\|\mathbf{v}\|(\hat{\mathbf{u}} \cdot \hat{\mathbf{v}})^\alpha$ and varying number of neurons. Training details are provided in the appendix.

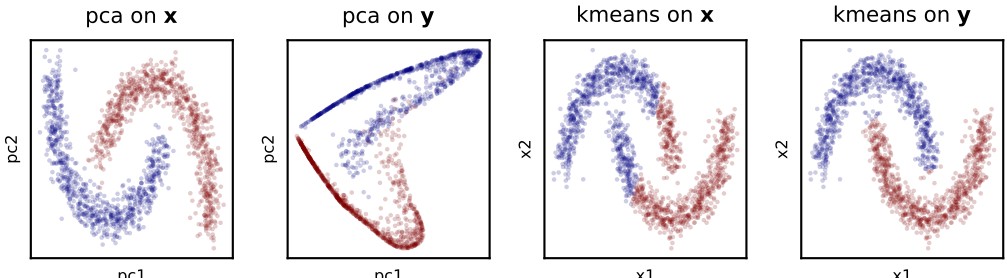

Figure 4: Utility of kernel similarity matching for downstream tasks. (a) principle components of the input vectors $\mathbf{x}$ (b) principle components of the 16 dimensional neural representations $\mathbf{y}$ (c) clustering generated by kmeans on $\mathbf{x}$ (d) clustering generated by kmeans on $\mathbf{y}$. For (a,b) the colors are given by ground truth labels while in (c,d) the colors are given by the KMeans clustering.

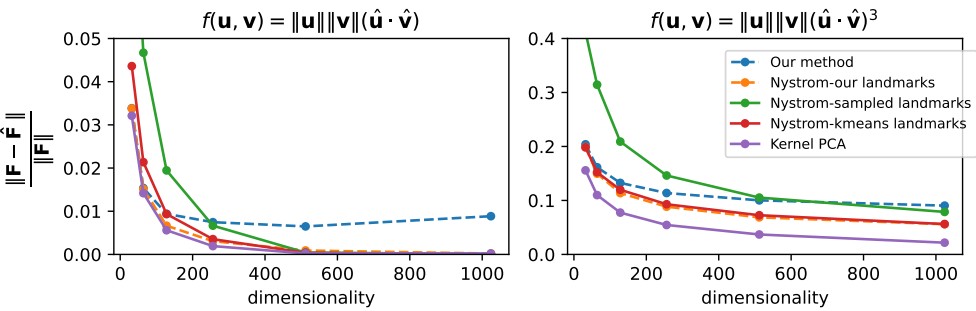

Figure 5: Approximation error vs. dimensionality for the MNIST dataset. (a) $f(\mathbf{u}, \mathbf{v}) = \mathbf{u} \cdot \mathbf{v}$ (linear kernel) (b) $f(\mathbf{u}, \mathbf{v}) = \|\mathbf{u}\|\|\mathbf{v}\|(\hat{\mathbf{u}} \cdot \hat{\mathbf{v}})^3$ (a nonlinear kernel). For the linear kernel all methods give relatively small approximation error once $n > 100$. Although yet again we see that the neural method does not continue to decrease as the dimensionality increases beyond 200, even in the linear setting.

The linear kernel is recovered by setting $\alpha = 1$. We are not aware of other works using this exact "power-cosine" kernel before, however it is motivated by the *arccosine kernel* studied in the context of wide random ReLU networks [Cho and Saul, 2009]. An important property of our kernel network is the linear input-output scaling, meaning that rescaling an input $\mathbf{x}' \leftarrow a\mathbf{x}$ will cause the corresponding representation to also be rescaled by the same factor $\mathbf{y}' \leftarrow a\mathbf{y}$. This will allow our nonlinear networks to have the same "contrast-invariant-tuning" properties that are thought to be displayed by simple cells in cat visual cortex [Skottun et al., 1987].

**Approximation error** We display the normalized approximation errors for $\alpha = 1$ and $\alpha = 3$ in Fig. 5. For the linear kernel ($\alpha = 1$) all methods yield a relatively small error even for low dimensionality. An error of $0.01$ is hard to see by eye when plotting input-output similarity scatter plots as done in Fig. 3. For both $\alpha = 1, 3$ we observe again a strange behavior of our method: it seems to "bottom out" and the error stops decreasing and even begins to increase as the dimensionality increases. This may be related to unstable convergence properties of gradient descent ascent.

For $\alpha = 3$ we observe that the kernel PCA method largely outperforms all methods. We observe that Nyström with either our features or KMeans appears to outperform sampled Nyström methods. The sampled Nyström method is worse than our representations for low dimensionality but eventually catches up and surpasses ours neural representations.

**Emergence of sparse representations** We train networks with $\alpha = \{1, 2, 3, 4\}$ and $n = 800$ neurons (so the output dimensionality is exactly 2x the input dimensionality). There is a sign degeneracy when $\alpha$ is odd: we can multiply both $\mathbf{w}_i$ and $y_i$ by $-1$ and the objective is unchanged. When we look at the response histogram for single neurons, we observe that for $\alpha = 3$, the response tends to be heavily skewed so that when the response has a high magnitude, it is either always positive or

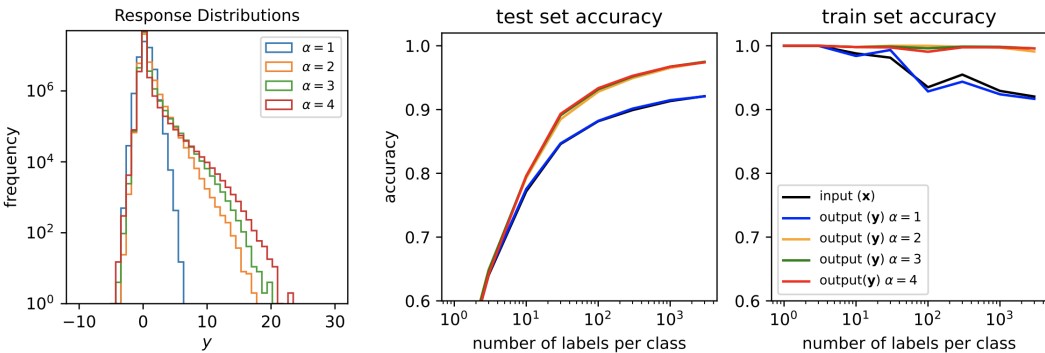

Figure 6: (a) response distribution (after the procedure we describe in the text for removing the sign degeneracy). The nonlinear kernels ($\alpha = 2, 3, 4$) naturally give rise to sparse distributions. (b) test set accuracy of a linear classifier classification for MNIST (c) train set accuracy of the corresponding linear classifier. Interestingly all nonlinear kernels give nearly identical train and test set results. The linear kernel gives nearly identical results to simply training the classifier directly on the pixels.

always negative. We remove this degeneracy by multiplying both $\mathbf{w}_i$ and $y_i$ by the sign of $\langle y_i \rangle$. After removing this degeneracy we plot the neuron responses over all patterns in Figure 6.

For $\alpha = 1$ (linear neurons), neuron responses are roughly centered around zero: neuron responses are neither sparse not skewed. For $\alpha = 2, 3, 4$, neurons appear to have a heavy tailed distribution, they frequently have small responses, but occasionally have large positive responses. Neurons become increasingly sparse and heavy tailed as we increase $\alpha$, although this effect is not that strong.

**Evaluating the representations via linear classification** We train a linear classifier on the inputs ($\mathbf{x}$) and the outputs ($\mathbf{y}$) for $\alpha = 1, 2, 3, 4$ and $n = 800$. We train every configuration using $k \in \{1, 3, 10, 30, 100, 300, 1000, 3000\}$ labels per class. We train all configurations with a weight decay parameter $\lambda \in \{1e-5, 1e-4, 1e-3, 1e-2, 1e-1, 1\}$ which yields the highest test accuracy. We average the accuracy for every configuration over 5 random seeds. The results are shown in Fig. 6.

As expected, the performance of the inputs and $\alpha = 1$ (linear similarity matching) is nearly identical on both test and train sets. Surprisingly, the test performance of $\alpha = 2, 3, 4$ is nearly identical. Perhaps these curves can be partially explained by the spectra of the output similarity matrix which we show in Figure **??** of the Appendix. While the shapes of the spectra are different in every case, $\alpha = 1$ has roughly 200 nonzero eigenvalues while $\alpha = 2, 3, 4$ all have nearly 800 nonzero eigenvalues. Perhaps the number of nonzero eigenvalues is more influential for the linear classification performance than the detailed shapes of these spectra.

## 5   Related work

**Kernel similarity matching with random Fourier features** The most closely related work to ours is kernel similarity matching with random Fourier features [Bahroun et al., 2017]. The key difference between our methods is that instead of learning the features $\mathbf{w}$, they use random Fourier features to directly generate nonlinear feature vectors $\phi^t = \sqrt{2/n} \cos(\mathbf{W}\mathbf{x}^t + \mathbf{b})$ which they then feed into a standard linear similarity matching network. This leads them to a different architecture (one feedforward layer + one recurrent layer, instead of our single recurrent layer net) and a different set of learning rules. A benefit of the random feature approach is that it will theoretically lead to perfect matching, so long as the number of random features is sufficiently large.

However, the feature learning aspect of our algorithm naturally led to a sparse set of responses which lends our model an added degree of biological plausibility. Additionally, our method generalizes to non-shift invariant kernels and empirically it yields better approximation error when the dimensionality of the output is sufficiently low. Our method can be seen as a biased method for approximation, which can be useful when the dimensionality is low, but ultimately will underperform compared to non-biased methods such as random Fourier methods or Nyström methods.

**Nyström Approximation** While not obviously biological, Nyström methods are perhaps the most commonly used methods for approximating kernel matrices. The Nyström approximation uses a set of landmarks $\{\mathbf{w}_i : i = 1, 2, \ldots, N\}$ to construct a low rank approximation of the original kernel matrix [Williams and Seeger, 2001]. To more clearly see the relationship between this method to ours, one can slightly modify the original formulation to generate "Nyström features":

$$\text{"Nyström features"} \quad y_j^t = \sum_i f(\mathbf{x}^t, \mathbf{w}_i) M_{ij} \text{ where } M_{ij} = [(\mathbf{B}^\dagger)^{1/2}]_{ij} \text{ and } B_{ij} = f(\mathbf{w}_i, \mathbf{w}_j) \tag{18}$$

$\mathbf{B}^\dagger$ indicates the pseudo-inverse. Multiplying two such vectors togethers yields the Nyström approximation $\hat{F}_{st} = \mathbf{y}^s \cdot \mathbf{y}^t = \sum_{ij} f(\mathbf{x}^s, \mathbf{w}_i)[\mathbf{B}^\dagger]_{ij} f(\mathbf{x}^t, \mathbf{w}_j)$. Our method produces representations of the same functional form but our $M$ matrix is derived from parameters learned by the correlations:

$$\text{Our features} \quad y_j^t = \sum_i f(\mathbf{x}^t, \mathbf{w}_i) M_{ij} \text{ where } M_{ij} = [\mathbf{L} + \lambda \mathbf{I}]_{ij}^{-1} q_j \tag{19}$$

As measured by squared error, the Nyström approximation was actually a better approximation than our representations, when we used the same set of landmarks (Figs. 3, 5). The variation in Nyström methods primarily come from the method used to generate the landmarks. Two broad categories of landmark selection can be defined: template vs. pseudo-landmark. Template based approaches choose landmarks as a subset of the inputs $\mathbf{w} \in \{\mathbf{x}_1, \mathbf{x}_w, \ldots, \mathbf{x}_T\}$ typically chosen via sampling schemes [Williams and Seeger, 2001, Drineas et al., 2005, Musco and Musco, 2017]. Pseudo-landmark approaches do not require the landmarks to be inputs. Zhang et al. [2008] used the cluster centers generated by KMeans as the landmarks. Fu [2014] formulate landmark selection as an optimization problem in the reproducing Hilbert space. Our method can be seen as a pseudo-template approach as our landmarks are directly generated via Hebbian learning rules and in general will not be exactly equal to any particular input pattern. Our method is similar in spirit to the approach of Fu [2014]. A key difference is that we use a different objective, a correlation-based upper bound to the sum of squared errors, which gives rise to correlation-based learning rules.

## 6 Discussion

We have extended the neural random Fourier feature method of Bahroun et al. [2017] for kernel similarity matching to instead be applicable to arbitrary differentiable kernels. Rather than using random nonlinear features, we learned the features with Hebbian learning rules. Both this work and that of Bahroun et al. [2017] can be seen as extensions of the linear similarity matching works written in Hu et al. [2014], Pehlevan and Chklovskii [2014], Pehlevan et al. [2015, 2018]. By using a nonlinear input similarity, the representations learned by our network are capable of learning high-dimensional nonlinear functions of the input, without requiring any constraints such as non-negativity.

To our knowledge this is the first work that attempts to directly optimize the sum of squared errors in Eq. 1 without relying on sampling schemes or direct computation of the input similarity matrix. It would be interesting to relax the correlation-based constraint we have imposed on ourselves. This might allow for a variety of different types of bounds (Eq. 3) to be derived which in turn could lead to more faithful approximations than the one presented in our paper.

Our work falls in line with an increasing body of literature that derives nonlinear Hebbian/local learning rules by starting with kernel/similarity matching measures Pogodin and Latham [2020], Nøkland and Eidnes [2019]. A key contribution of our work is the use of an unsupervised objective, rather than a supervised objective, and a purely bottom up flow of information from pixels to features.

The computational complexity of our method (after learning) is comparable to Nyström methods. Let d be the input dimensionality and D be the feature dimensionality. Then Nyström methods require have computational complexity of $Dd + D^3$ and random Fourier methods only require require $Dd$. Like Nyström, our proposed method requires $Dd + D^3$ if using the fast inverse method (eqn 12). Implementation of the proposed method with a recurrent network (Eq. 11) requires $Dd + D^3 K$ where $K$ is number of recurrent network iterations.

A key obstacle faced by users of this algorithm is the stochastic gradient descent-ascent procedure. Empirically the convergence of our algorithm is quite sensitive to the learning rates. This method does not provide the same sorts of theoretical guarantees or empirically observed robustness of sampling

based methods. Use of more robust descent-ascent optimization methods could be useful for making this class of algorithms more accessible for the practitioner.

## Acknowledgements and Funding

We would like to thank Lawrence Saul and Runzhe Yang for their helpful insights and discussions.

This research was supported by the Intelligence Advanced Research Projects Activity (IARPA) via Department of Interior/ Interior Business Center (DoI/IBC) contract number D16PC0005, NIH/NIMH RF1MH117815, RF1MH123400. The U.S. Government is authorized to reproduce and distribute reprints for Governmental purposes notwithstanding any copyright annotation thereon. Disclaimer: The views and conclusions contained herein are those of the authors and should not be interpreted as necessarily representing the official policies or endorsements, either expressed or implied, of IARPA, DoI/IBC, or the U.S. Government.

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
