# Appendix

## 1 Interpretation using rank-1 Nyström approximation

The bound in Equation 5 of the main paper can be interpreted using a rank-1 Nyström approximation for $f(\mathbf{x}_t, \mathbf{x}_{t'})$. By holding $\mathbf{w}$ fixed and maximizing for $q$ in the right hand side of Equation 5, we get $q^* = f(\mathbf{w}, \mathbf{w})^\dagger \sum_t y_t f(\mathbf{x}_t, \mathbf{w})$ where $f(\mathbf{w}, \mathbf{w})^\dagger$ indicates the pseudo-inverse.[1] We can insert this optimal $q^*$ back into the right hand side to yield:

$$\sum_t q^* y_t f(\mathbf{x}_t, \mathbf{w}) - \frac{1}{2}(q^*)^2 f(\mathbf{w}, \mathbf{w}) = \frac{1}{2}\sum_{t,t'} y_t f_{t,t'}^{\text{Nyström}} y_{t'} \tag{1}$$

where we have defined they Nyström matrix:

$$f_{t,t'}^{\text{Nyström}} := f(\mathbf{x}_t, \mathbf{w})f(\mathbf{w}, \mathbf{w})^\dagger f(\mathbf{w}, \mathbf{x}_{t'}) \tag{2}$$

The matrix $f_{t,t'}^{\text{Nyström}}$ is a rank-1 Nyström approximation for the full similarity matrix $f(\mathbf{x}_t, \mathbf{x}_{t'})$ Williams and Seeger [2001]. Note that for every dimension $i$ of the representation vector $\mathbf{y}$, we have a different landmark vector $\mathbf{w}^i$ so we are using a different rank-1 approximation of the matrix $f(\mathbf{x}_t, \mathbf{x}_{t'})$ for every to the pairwise sum $\frac{1}{2}\sum_{t,t'} y_t^i y_{t'}^i f(\mathbf{x}_t, \mathbf{x}_{t'})$.

Typically the weight vector $\mathbf{w}$ , often called a "landmark", used in the Nyström approximation is set either by setting it to a random input or by more sophisticated schemes like setting it with KMeans. In our case, we are directly optimizing the landmarks via Equation 6 in the main paper. To our knowledge the only other work to do this was performed in Fu [2014].

## 2 PyTorch code for training

The code used in the main training loop of our algorithm is shown in Fig. 1.

## 3 Homogeneous kernels

Before moving on we note that a simplification can be made if we have a homogenous (scale free) kernel, i.e. if $f(a\mathbf{u}, b\mathbf{v}) = (ab)^\alpha f(\mathbf{u}, \mathbf{v})$. Examples of such kernels are the linear kernel $f(\mathbf{u}, \mathbf{v}) = \mathbf{u} \cdot \mathbf{v}$, homogeneous polynomial kernels $f(\mathbf{u}, \mathbf{v}) = (\mathbf{u} \cdot \mathbf{v})^\alpha$, and the cosine-based kernel we will use in one of our experiments $f(\mathbf{u}, \mathbf{v}) = \|\mathbf{y}\|\|\mathbf{v}\|(\hat{\mathbf{u}} \cdot \hat{\mathbf{v}})^\alpha$. In this case, there is a degeneracy between $q_i$ and the norm of $\mathbf{w}_i$. This means we can actually eliminate the minimization over $\mathbf{q}$ by setting $q_i = 1$. We prove this fact in the appendix. In the case of a homogeneous kernel, we are left with the simpler equivalent optimization:

$$\min_{\mathbf{W}} \max_{\mathbf{L}} \min_{\mathbf{Y}} -\sum_{i=1}^{N}\left[\langle y_i f(\mathbf{w}_i, \mathbf{x})\rangle - \frac{1}{2}f(\mathbf{w}_i, \mathbf{w}_i)\right] + \frac{1}{2}\sum_{i,j=1}^{N}\left[L_{ij}\langle y_i y_j\rangle - \frac{1}{2}L_{ij}^2\right] + \lambda\langle y_i^2\rangle \tag{3}$$

This more clearly shows the relationship between the linear similarity matching objectives and the more general kernel similarity matching objective. When $f(\mathbf{w}_i, \mathbf{x}) = \mathbf{w}_i \cdot \mathbf{x}$, we are in fact left with

---

[1]The pseudo-inverse of the scalar $f(\mathbf{w}, \mathbf{w})$ acts exactly like the regular inverse except it is defined to be 0 when $f(\mathbf{w}, \mathbf{w})$ is zero, unlike the regular inverse which would be undefined.

36th Conference on Neural Information Processing Systems (NeurIPS 2022).

```python
# train loop
for i in range(n_iter):
    # inference
    x = next(loader)
    y = self.forward(x)

    # gradients
    e = self.energy(x,y)
    gq, gw, gl = torch.autograd.grad(e, [self.q, self.w, self.l])

    # updates
    with torch.no_grad():
        self.q += etaq * gq
        self.w += etaw * gw / (self.q**2).unsqueeze(1)
        self.l -= etal * gl
```

Figure 1: Training loop to perform the GDA-based optimazation of Eq. 9 in the main paper written using PyTorch

the exact objective studied in previous works on linear similarity matching Pehlevan et al. [2018]. This simplification can be easily implemented in code by initializing $q_i = 1$ and setting the learning rate for $\eta_q$ to be 0 for all iterations.

## 4   Proof that the bound in Equation 6 is maximized when $q = 1$ and $f$ is a homogeneous kernel

Assume that $f$ is a homogenous kernel, so that $f(\lambda_1 \mathbf{u}, \lambda_2 \mathbf{v}) = (\lambda_1 \lambda_2)^\alpha f(\mathbf{u}, \mathbf{v})$ for any $\lambda_1, \lambda_2 > 0$. We will show that in this case we can simply set $q = 1$. Assume we have some pair $q, \mathbf{w}$. Then define $q' = 1$ and $\mathbf{w}' := q^{1/\alpha} \mathbf{w}$. Because our kernel is homogeneous, we have $f(\mathbf{w}', \mathbf{x}_t) = f(q^{1/\alpha} \mathbf{w}, \mathbf{x}_t) = q f(\mathbf{w}, \mathbf{x}_t)$ and similarly $f(\mathbf{w}', \mathbf{w}') = q^2 f(\mathbf{w}, \mathbf{w})$. In other words when we have a homogenous kernel, we can always just rescale the features $\mathbf{w}' \leftarrow q^{1/\alpha} \mathbf{w}$ so the following holds for any $q$:

$$\sum_t q y_t f(\mathbf{x}_t, \mathbf{w}) - \frac{1}{2} q^2 f(\mathbf{w}, \mathbf{w}) = \sum_t y_t f(\mathbf{x}_t, \mathbf{w}') - \frac{1}{2} f(\mathbf{w}', \mathbf{w}') \tag{4}$$

## 5   Methods we compare to in our experiments

**Kernel PCA** The optimal (in terms of mean squared error) rank $N$ approximation $\hat{f}$ to the kernel matrix $f$ is given by the top $n$-dimensional subspace of the kernel matrix [Borg and Groenen, 2005]. Specifically, we perform an eigenvector decomposition on $f$ then set $\hat{f}$ via:

$$f(\mathbf{x}_s, \mathbf{x}_t) = \sum_{i=1}^{T} \lambda_i \mathbf{v}_i \mathbf{v}_i^\top \rightarrow \hat{f}_{st} = \sum_{i=1}^{N} \lambda_i \mathbf{v}_i \mathbf{v}_i^\top \tag{5}$$

For the mnist dataset, the kernel matrix is 70k x 70k entries so we use a randomized svd algorithm to compute the top components, rather than a full SVD. We use the PyTorch implementation "torch.svd_lowrank" with $q = 1024 + 256$ (so we estimate the top 1024+256 singular values and vectors) and we set niter=4 meaning we do 4 power iterations.

**Nystrom methods** Given a set of landmarks $\{\mathbf{w}_i : i = 1, 2, \ldots, N\}$, the nystrom method defines two matrices:

$$A_{ti} = f(\mathbf{x}^t, \mathbf{w}_i) \qquad B_{ij} = f(\mathbf{x}_i, \mathbf{w}_j) \tag{6}$$

These are used to approximate the kernel matrix via:

$$\hat{f}_{st} = \sum_{ij} A_{si} [\mathbf{B}^\dagger]_{ij} A_{tj}^\top \tag{7}$$

To calculate the pseudo-inverse of $\mathbf{B}$ we use double precision arithmetic and set first set all singular value of $B$ smaller than $1e - 10$ to zero. We compare 3 different methods of landmark generation in our paper.

**Nystrom with uniformly sampled landmarks** This is the simplest method, and was proposed in he original paper using the Nystrom method to approximate kernel matrices [Williams and Seeger, 2001]. We simply uniformly sample $N$ landmarks without replacement from the dataset.

**Nystrom with landmarks generated via KMeans** This method was used by Zhang et al. [2008] and instead uses the cluster centers given by KMeans as the landmarks. We initialize our means with templates from the dataset and the use Lloyd's method to update our cluster centers [Lloyd, 1982]. This is run either until convergence, or 100 iterations of the algorithm occurs, whichever happens first.

**Nystrom with landmarks generated via our method** We use the $N$ features learned with Hebbian update rules as the landmarks in the Nystrom approximation.

**Random Fourier features** For the half moons dataset using the Gaussian kernel, we also compare our method to the random Fourier feature method [Rahimi et al., 2007]. The authors in Bahroun et al. [2017] train a linear similarity matching on top of these features. But for simplicity, we just use the random features themselves, rather than the subsequent neurally generated features. This provides a best-case scenario for the neural random Fourier method. The neural algorithm is simply trying to matching similarities $\min y \|\mathbf{y}^s \cdot \mathbf{y}^t - \boldsymbol{\phi}^{\boldsymbol{s}} \cdot \boldsymbol{\phi}^{\boldsymbol{t}}\|$, and it should be able to provide zero error, given the same output dimension as input dimension. Although it practice it can be challenging to set the learning rates appropriately, so we evaluate $\phi$ instead of $\mathbf{y}$ to avoid any possible issues with improper training.

To generate these features, we randomly sample $\mathbf{w}_i \sim \mathcal{N}(\text{mean} = 0, \text{variance} = \frac{1}{\sigma^2}\mathbf{I})$ where and $b_i \in \text{Uniform}[0, 2\pi]$ as set the features as

$$\phi_i^t = \sqrt{\frac{2}{n}} \cos(\mathbf{w}_i \cdot \mathbf{x}^t + b_i) \tag{8}$$

# 6 Half Moons Experiment

### 6.0.1 Training details

We train with minibatch sizes of 64 input. We train for 10000 iterations with $\eta_w = \eta_q = 0.01$ and $\eta_l = 0.1$. Then we anneal the learning rates by a factor of 10x and train for 10000 more iterations $\eta_w = \eta_q = 0.001$ and $\eta_l = 0.01$.

# 7 MNIST Experiment

## 7.1 Training details

We train with minibatch sizes of 64. After initialization and warmup, we set $\alpha$ and train for 10,000 iterations with $\eta_w = 0.001, \eta_l = 0.01$. We then decay the learning rates for $W, L$ by 10x and train for with 5k more iterations with $\eta_w = 0.0001, \eta_l = 0.001$. This whole procedure takes approximately 4 minutes on an NVIDIA GTX 1080 GPU.

## 7.2 Learned features for MNIST dataset

In Figure 2 we show the weights $\mathbf{w}_i$, visualized as 20x20 images, that are learned. When $\alpha = 1$ (linear similarity matching), the features appear as complicated linear combinations of input digits. However, with $\alpha = 2$ we see clear digits beginning to emerge. And with $\alpha = 4$ nearly all the features look like whole digits.

# 8 Receptive field analysis (aka "linearized neuron responses")

A natural way to visualize what the networks learn is to examine the feedforward weights. However these visualizations are not as interpretable in this experiment as they were for the simple halfmoons

## Feedforward weights

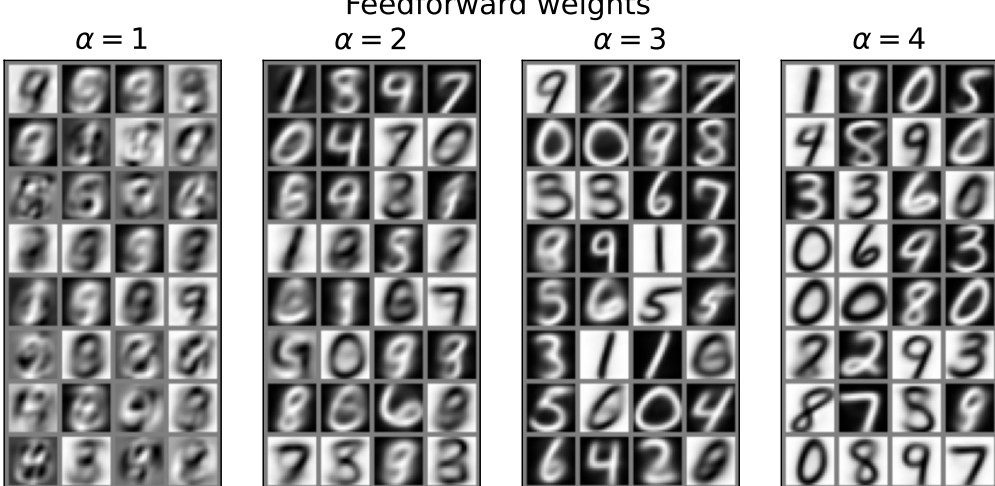

Figure 2: Feedforward weights ($\mathbf{W}$) learned by the network for $\alpha = 1, 2, 3, 4$. When $\alpha = 1$, the weights appear to be complicated linear combinations of input vectors. As $\alpha$ increases, the weights begin to resemble "templates", i.e. whole digits. In the main text, we argue this behavior results from the increasing sharpness of neural tuning as $\alpha$ increases.

## Linearized Neuron Responses

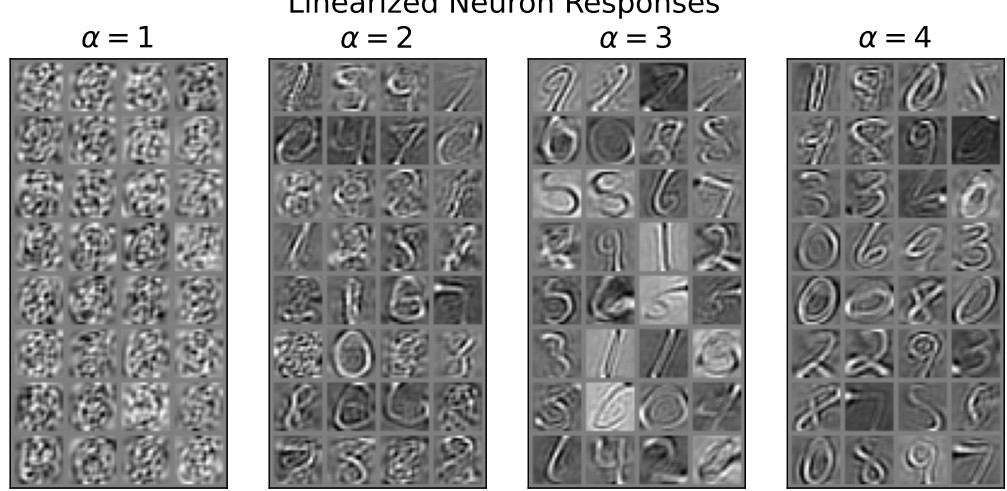

Figure 3: "Linearized responses" for a subset of neurons from networks with $\alpha = \{1, 2, 3, 4\}$. Specifically, for each neuron $y_i$ we compute the vector $\mathbf{s}_i = \left[ 0.1\,\mathbf{I} + \langle \mathbf{x}\mathbf{x}^\top \rangle \right]^{-1} \langle y_i \mathbf{x} \rangle$ and visualize $\mathbf{s}_i$ as a $20 \times 20$ image. As $\alpha$ increases, it appears that neurons become increasingly selective to whole input digits.

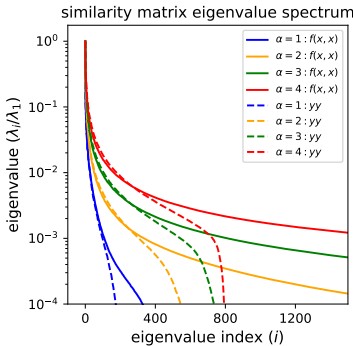

Figure 4: Eigenvalue spectrum of the input similarity matrix $f(\mathbf{x}_t, \mathbf{x}_{t'})$ and learned output similarity matrix $\mathbf{y}_t \cdot \mathbf{y}_{t'}$. If similarity matching were optimal, (i.e. we just performed uncentered kernel PCA on the input similarity matrix) the largest 800 eigenvalues would be exactly matched and subsequent eigenvalues would be zero. We see that increasing $\alpha$ brings up the tails of the spectrum, approximately "whitening" the responses. For $\alpha = 1$, because the inputs are 400 dimensional, the spectrum only has at most 400 nonzero eigenvalues.

dataset. In particular for $\alpha = 1, 2, 3$ the weights appear to be a blend of templates (whole digits) and more complicated linear combinations of digits. We show some examples from each network configuration in the appendix.

We can better understand and visualize the network responses by instead examining the linearized neuron responses. Specifically, for each neuron $y_i$ we compute the vector $\mathbf{s}_i = \left[ 0.1\,\mathbf{I} + \langle \mathbf{x}\mathbf{x}^\top \rangle \right]^{-1} \langle y_i \mathbf{x} \rangle$. This vector can be thought of as a linear approximation to each neuron $y_i \approx \mathbf{s}_i \cdot \mathbf{x}$. We show these vectors, again visualized as images, in Figure 3.

These linearized responses actually highlight a behavior not seen by only considering feedforward weights. We see for $\alpha = 2$, it appears that many of the neurons appear to be selective for smaller regions of the input, sometimes interpretable as strokes and edges. This behavior is likely coming from some sort of cancellation between the feedforward input and lateral interactions. As $\alpha$ increases, the linear filters appear to grow in size to resemble whole digits.

For $\alpha = 1$ (aka linear similarity matching) the linearized responses do not in any way appear as whole digits, rather they appear to be high spatial frequency images. This is not a failure of the networks, as the input-output similarities are nearly perfectly matched. This behavior results from the fact that linearity is not enough to encourage parts or whole templates to be learned.

## 8.1 Spectral analysis of the representations

We examine the eigenvalue spectrum of the input similarity matrix $f(\mathbf{x}_t, \mathbf{x}_{t'})$ and the output similarity matrix $\mathbf{y}_t \cdot \mathbf{y}_{t'}$. We plot these spectra in Figure 4. Note that we normalize the spectra by dividing by the largest eigenvalue.

Without even considering the output representations, we can already observe interesting behavior just by considering the spectrum of the input similarity matrix. As we increase $\alpha$, the "sharpness" of the kernel, the spectrum of $f$ tends to flatten out. The effective rank of this matrix increases with increasing kernel sharpness. This observation is is in part a motivation for kernel similarity matching. Matching a high rank matrix naturally requires high dimensional vectors. This increase in dimensionality may be useful for downstream tasks such as linear classification. It is also an important part of brain inspired modeling to use overcomplete representations of the input Olshausen and Field [1997].

For $\alpha = 1$, the spectrum of $\mathbf{y}_t \cdot \mathbf{y}_{t'}$ closely matches the spectrum $f(\mathbf{x}_t, \mathbf{x}_{t'})$ for the larger eigenvalues. However, it appears to fall off for smaller eigenvalues. This may be due in part to the training not being fully converged. For $\alpha > 1$, the spectrum of $\mathbf{y}_t \cdot \mathbf{y}_{t'}$ approximately matches for the larger eigenvalues (although not perfectly). However again the spectrum tends to fall off more rapidly for the learned representations than for the input similarity matrix. Note that because the dimensionality

of $\mathbf{y}$ is 800 for all experiments, the spectrum necessarily must be zero for all eigenvalues smaller than the 800th largest eigenvalue.