# OpenReview forum: "Kernel similarity matching with Hebbian networks"
_NeurIPS.cc/2022/Conference — NeurIPS 2022 Accept_

### Official Review · Reviewer_2oLH · 2022-07-06

**Rating:** 6
**Confidence:** 3
**Soundness:** 3 good
**Presentation:** 3 good
**Contribution:** 1 poor

**Summary:**

The paper proposes an on-line kernel similarity matching algorithm.
Taking inspiration from biology, the authors wish to derive a representation learning algorithm that does not consider pairwise similarities (or kernel matrices), and that is based on online parameter updates.
They take the classical multidimensional scaling objective, and derive an upper bound that can be computed without pairwise terms, and further modify it to allow for online computation.
Learning rule examples are provided for a few kernels.
An experimental evaluation on toy data and MNIST illustrates that the method works similar to kernel PCA in low dimensional settings, but less good on more challenging settings.

EDIT AFTER REBUTTAL: I have increased my initial score from 4 to 6 after reading the other reviews and discussing a few of my questions with the authors.

**Questions:**

It does not seem very clear to me what the high level goal of this paper is. A new method that performs well, a method that leads to interesting insights?

**Limitations:**

Numerical limitations (stability) are addressed.
Societal impact not applicable.

**Strengths And Weaknesses:**

Caveat: I am not very well read in the Hebbian learning literature.

A well written paper that proposes an interesting idea.
The upper bound reconstruction and modifications for online learning are clever.
The experiments are on very limited and low dimensional datasets and do not include any modern baselines for representation learning (only kernel pca / kmeans).
The main premise of the paper is that something can be learned by deriving brain-inspired learning algorithms, yet the paper does not deliver on this premise: the results are not very impressive and neither are their any take-aways from deriving this methodology.

In its current form, it does not provide enough content, or impact to be published as a conference paper at Neurips. I would suggest either a workshop submission or a resubmission with more interesting applications, results, or insights.

---

> ### Author Response · Authors · 2022-08-02
> **Reply to reviewer 2oLH**
>
> *It does not seem very clear to me what the high level goal of this paper is. A new method that performs well, a method that leads to interesting insights?*
>
> The high level goal is to address a fundamental theoretical question about the brain: what is the computational function of Hebbian synaptic plasticity? This biological phenomenon has been studied extensively, but its computational interpretation remains unclear. According to a theory of Chklovskii and colleagues, Hebbian plasticity enables neural nets to learn representations by the principle of similarity matching. But merely preserving similarities does not seem powerful enough to explain learning in the brain. Instead we propose that Hebbian neural nets transform similarities according to a kernel function that is closely related to the nonlinear activation function of the neurons. In short, the paper is about brain theory. We are not claiming that our method will compete with state-of-the-art methods in practical applications, but we do show empirically that its performance is comparable to or exceeds that of closely related methods.
>
> Recent works on linear similarity matching
>  - https://direct.mit.edu/neco/article-abstract/30/1/84/8348/Why-Do-Similarity-Matching-Objectives-Lead-to?redirectedFrom=fulltext
>  - https://proceedings.neurips.cc/paper/2019/hash/222afbe0d68c61de60374b96f1d86715-Abstract.html
>  - https://arxiv.org/abs/1503.00669?context=cs

---

> > ### Comment · Reviewer_2oLH · 2022-08-03
> > **Thank you for the reply**
> >
> > Many thanks for the references and the additional context. I appreciate I am outside of this community, there are other/similar papers with similar ideas published, and the other reviews are generally favorable. I am happy to revise my score to that consensus, give that the paper is well written in general.
> >
> > However, I have to say that I am still puzzled what the take-away from the proposed methodology is. A biologically inspired online kernel matching algorithm performs reasonably on toy data, but what do we learn from this, about the brain or about computation? Your explanations above seem to motivate studying the questions at hand, but not what we learn or gain from that.

---

> > > ### Author Response · Authors · 2022-08-09
> > > **Response**
> > >
> > > Thanks!
> > >
> > > As to the motivation, one might imagine this work as trying to identify "lego blocks" of neural computation. Can we identify basic building blocks that are biologically plausible, that we can then piece together, perhaps hierarchically, into a useful computational device? We hope that this paper provides one lego block, nonlinear similarity matching, that can hopefully be either stacked into a hierarchy or combined with other mechanisms such as pooling/attention mechanisms, to generate useful visual representations.

---

### Official Review · Reviewer_3WSb · 2022-07-10

**Rating:** 6
**Confidence:** 2
**Soundness:** 3 good
**Presentation:** 3 good
**Contribution:** 3 good

**Summary:**

Extended neural random Fourier feature method of Bahroun et al. [2017] for kernel
similarity matching being applicable to arbitrary differentiable kernels. Rather than using
random nonlinear features the features are learned with Hebbian learning rules.
More or less an online method for learning a (psd) similarity matrix.


**Questions:**

- simple question for the beginning of the paper why is
  'neural kernel similarity matching' relevant?
- Your objective is interesting (but should be a bit more motivated and clarified already in
  sec 1
  (Side note regarding 'online' -- there is also work on online SVD / EVD and online nystroem
   see e.g https://arxiv.org/pdf/1802.07887.pdf)
- Regarding Eq 1 if T is super large your kernel matrix could be large as well
  ... given we make a SVD or EVD on the (psd) kernel matrix we can have your y simple
  by making an euclidean embedding of your kernel ... - which could be done more
  or less reasonable efficient e.g. by Nystroem approximation
  See e.g. work of Talwalkar https://www.jmlr.org/papers/v14/talwalkar13a.html
- 'This is known as the classical multidimensional scaling objective' - well yes but
  for MDS the embedding dimension is typically very low dimensional
- has the kernel function to be shift invariant (constant on the diagonal)?
- derivations appear to be ok
- hm, convergence is obviously not easy
- Fig 3 there are dashed lines in the plot but not (?) in the legend - maybe due to scaling issues?
- little typo 'psuedo-inverse'
- please replace arxiv links by regular publications if available e.g.
  Cameron Musco, Christopher Musco: Recursive Sampling for the Nystrom Method. NIPS 2017: 3833-3845


**Limitations:**

does not apply

**Strengths And Weaknesses:**

Strength:
- given the approach is really more cost effective in balance to accuracy it is interesting
  to have an online version for the learning of similarities ~ kernel function approximation
- nice theoretical derivation which maybe of interest on its on (motivated from neuro-biology)

Weakness:
- convergence is obviously a problem - it may not be clear if the approach scales always
  so nice to a wider set of benchmark data

---

> ### Author Response · Authors · 2022-08-02
> **Reply to reviewer 3WSb**
>
> *convergence is obviously a problem*
>
> We hope that the theoretical derivation of the neural network is interesting enough on its own to warrant publication. The issue of convergence may be addressed in future work.
>
> *simple question for the beginning of the paper why is 'neural kernel similarity matching' relevant?*
>
> There is a great deal of biological evidence for Hebbian synaptic plasticity in the brain. However, the computational purpose served by Hebbian plasticity remains unclear. According to a theory of Chklovskii and colleagues, Hebbian plasticity enables neural nets to learn representations by the principle of similarity matching. After learning, the similarities between the output vectors of such a net match the similarities between the input vectors. This is an important advance for brain theory. However, simply matching similarities seems computationally limited. The principle does not seem powerful enough to explain learning by the brain. Similarities between representation vectors in the visual system, for example, do not stay the same as one moves higher in the visual hierarchy, but are transformed to better support invariant object recognition. Therefore it is natural to ask whether Hebbian plasticity can support transformation rather than matching of similarities. Kernel similarity matching can be regarded as a systematic transformation of similarities by a nonlinear kernel function.
>
>
> *Your objective is interesting (but should be a bit more motivated and clarified already in sec 1 (Side note regarding 'online' -- there is also work on online SVD / EVD and online nystroem see e.g https://arxiv.org/pdf/1802.07887.pdf)*
>
> Thanks for the reference. We should clarify that the goal of the paper is a neural algorithm, which should not only be online but also employ local correlation-based rules of synaptic plasticity. A neural algorithm is by definition online, but the converse is not necessarily true.
>
> *Regarding Eq 1 if T is super large your kernel matrix could be large as well ... given we make a SVD or EVD on the (psd) kernel matrix we can have your y simple by making an euclidean embedding of your kernel ... - which could be done more or less reasonable efficient e.g. by Nystroem approximation See e.g. work of Talwalkar https://www.jmlr.org/papers/v14/talwalkar13a.html*
>
> Correct, as the number of features grows, no learning is needed.
>
> *'This is known as the classical multidimensional scaling objective' - well yes but for MDS the embedding dimension is typically very low dimensional*
>
> Yes we will clarify that the objective function is familiar from MDS but we are interested in high-dimensional embeddings, which is not typical in MDS.
>
> *hm, convergence is obviously not easy*
>
> Since we have no theoretical guarantee of convergence, we have provided some numerical experiments showing that the method can converge to a reasonable solution in certain cases.
>
> *Fig 3 there are dashed lines in the plot but not (?) in the legend - maybe due to scaling issues?*
>
> Thanks for the identification. We will fix this.
>
> *little typo 'psuedo-inverse'*
> Thanks for the identification, we will fix this.
>
> *please replace arxiv links by regular publications if available e.g. Cameron Musco, Christopher Musco: Recursive Sampling for the Nystrom Method. NIPS 2017: 3833-3845*
>
> We will fix this

---

### Official Review · Reviewer_P8xq · 2022-07-11

**Rating:** 7
**Confidence:** 5
**Soundness:** 3 good
**Presentation:** 4 excellent
**Contribution:** 3 good

**Summary:**

The paper proposes a way to approximate a kernel using a similarity matching objective. In particular, the method adapts the tools from linear similarity matching that formulate kernel approximations as neural network training.

**Questions:**

### Comparison with other methods
The authors should include the standard random Fourier features (RFF), unless Bahroun et al. [2017] implementation is exact. Comparing with orthogonal random features [Yu et al, 2016] (https://proceedings.neurips.cc/paper/2016/hash/53adaf494dc89ef7196d73636eb2451b-Abstract.html) could be interesting as well, as it should improve performance of RFFs wrt dimensionality.

I'd also like to see comparison of computational complexity of Nystrom, RFF, neural RFF and the proposed method.

On a related note, similarity matching and kernel-based objective have been used for neural network training: [Nøkland and Eidnes, 2019] https://proceedings.mlr.press/v97/nokland19a.html (similarity matching), [Pogodin and Latham, 2020] https://proceedings.neurips.cc/paper/2020/hash/517f24c02e620d5a4dac1db388664a63-Abstract.html (kernels + nearly similarity matching). These methods are not directly comparable with the presented one, but maybe still worth discussing.

### Small corrections

Appendix has to be in a separate pdf.

Appendix E refers to a bound in Eq. 6, but there's no bound. I assume it's for Eq. 3. Either way, why is it an important result?

Fig. 3 should mention what kernel is used.

Line 83: Reference Eq. 3 explicitly to avoid confusion with Eq. 5.

Lines 4-5: unfinished sentence. “In this paper attempt” -> “This paper attempts” would fix it

Line 46: “we” tackle

Lines 256-258: “it seemed that” -> “it seemed” to fix the sentence

Line 295: descent-ascent? (add hyphen)

Line 296: convergence “of”

Line 297: theoretical (delet -ly)


**Limitations:**

The authors adequately addressed the limitations and potential negative societal impact.

**Strengths And Weaknesses:**

### Strengths
1. The proposed method is interesting as a biological implementation of kernel approximation, as it can work with one point at a time (and not pairs of points) to approximate the similarity matrix.
2. This is a relatively straightforward extension of the previous work on online similarity matching, but it's still novel.

### Weaknesses
1. Unlike Nystrom and random Fourier features method, the proposed method doesn't have theoretical guarantees for the quality of the approximation.
2. The paper is unlikely to be interesting outside of the neuroscience community, as random Fourier features (along with some improved methods) work well/fast/with guarantees for kernel approximation.


### Summary
I think it's an interesting paper with some room for improvement; 7 (accept).

---

> ### Author Response · Authors · 2022-08-02
> **Reply to reviewer P8xq**
>
> *The authors should include the standard random Fourier features (RFF), unless Bahroun et al. [2017] implementation is exact.*
>
> The neural random Fourier features are in theory exact, so we do not distinguish between the neural and non-neural features. The orthogonal features are interesting and we can try to add this to the half moons dataset.
>
> *I'd also like to see comparison of computational complexity of Nystrom, RFF, neural RFF and the proposed method.*
>
> Let d = input dimensionality and D = feature dimensionality. Then Nystrom requires Dd+D^3 and RFF requires Dd. The proposed method requires Dd+D^3 (identical to Nystrom) if using the fast inverse method (eqn 12). Implementation of the proposed method with a recurrent network (Eq. 11) requires Dd + D^3 K where K is number of iterations. We can add a brief note on the computational complexity.
>
> *Appendix E refers to a bound in Eq. 6, but there's no bound. I assume it's for Eq. 3.*
>
> This should have said the optimization, not the bound, in Eq. 6. This is important because in section 4, we use this trick, setting q = 1, to train the MNIST nets.
>
> And thanks for the other grammatical corrections. We will fix these in the final version.

---

> > ### Comment · Reviewer_P8xq · 2022-08-06
> > **Response to the authors**
> >
> > Thank you for the reply!
> >
> > I think you should clarify that neural RFF are performing exactly RFF in the text to avoid confusion; I also think the similarity matching papers linked above are relevant enough to this work. Otherwise, I'm happy with the response and I'm favouring acceptance (same score of 7).

---

> > > ### Author Response · Authors · 2022-08-09
> > > **Response**
> > >
> > > Thanks!
> > >
> > > We can clarify that the neural RFFs are exact. Additionally in the final draft we can provide some discussion/context for the more recent papers you linked.

---

### Official Review · Reviewer_rqwG · 2022-07-11

**Rating:** 6
**Confidence:** 2
**Soundness:** 3 good
**Presentation:** 3 good
**Contribution:** 3 good

**Summary:**


This paper proposes a neural random Fourier feature method. Its novelty is that it considers arbitrary differentiable kernels, while the method introduced by Bahroun et al. (2017) considered kernel similarity matching.


**Questions:**

The main question is about the incremental contribution of this paper, compared to the work of Bahroun et al. (2017)

**Limitations:**

yes

**Strengths And Weaknesses:**

The paper is well written and the contributions seem clear. The derivations seem novel and the experiments are interesting, illustrating the relevance of the proposed method

One issue is that this paper seems to provide an incremental contribution; This contributions is simply the extension of the neural random Fourier feature method be applicable to arbitrary differentiable kernels, while the initial method proposed by Bahroun et al. (2017) investigated kernel similarity matching.

There are some spelling and grammatical errors that can be easily identified and corrected, such as "the neural algorithm procedes as follows”, “We compare various number of neurons“, “the cluster centers return by KMeans”, “has occured”, "where the the neurons”, “the the colors are”, “We train all configuration”, “The results are show in”, “it seemed that to yield”, “Empirically the convergence our algorithm is”… And also several errors in the appendix.

---

> ### Author Response · Authors · 2022-08-02
> **Reply to reviewer rqwG**
>
> *There are some spelling and grammatical errors...*
>
> Thanks for the catches! We will fix all these in the final revision.
>
> *The main question is about the incremental contribution of this paper, compared to the work of Bahroun et al. (2017)*
>
> The problem of “kernel similarity matching” was previously studied in research on kernel PCA and non-metric multidimensional scaling. Output similarities (defined by inner product) are matched to input similarities (defined by a kernel function). The term “transformation” might be more apt than “matching,” because similarities are effectively transformed by the kernel function. This should be more powerful than regular similarity matching, which simply attempts to preserve similarities defined by inner product.
> Bahroun et al. (2017) were the first to propose a neural solution to kernel similarity matching. They use random Fourier features as inputs to an existing linear similarity matching algorithm. Their algorithm can be viewed as a two layer network. The first layer is feedforward, nonlinear, and does not learn. The second layer is recurrent, semilinear, and learns by (anti-)Hebbian plasticity.
> We propose an alternative neural solution to kernel similarity matching, which differs greatly from Bahroun et al. (2017). Our solution is a single layer network that is (anti-)Hebbian, recurrent, and nonlinear. The learning gives rise to features that resemble the input patterns, rather than Fourier features.

---

> > ### Comment · Reviewer_rqwG · 2022-08-08
> > **On the Hebbian/anti-Hebbian learning**
> >
> > I thank the authors for their reply.
> >
> > Unfortunately, the reply raised other issues within comparing the submitted paper to the work of Bahroun et al. (2017). In the latter, there is a clear presentation of the Hebbian/anti-Hebbian online learning rules. This is not the case in the current submitted paper. Of particular interest is that the submitted paper focuses on Hebbian learning in almost all the paper, while the authors in the reply focus on the anti-Hebbian nature of the proposed method.

---

> > > ### Author Response · Authors · 2022-08-09
> > > **Reply to on the Hebbian/anti-Hebbian learning**
> > >
> > > I believe I see the confusion. In our reply we distinguish between hebbian and anti-hebbian learning. This distinction is not made in the paper as we call every learning rule that involves local updates "Hebbian". Unfortunately in the literature, there is no clearly defined meaning of Hebbian as the original Hebb's postulate was not a mathematical formula. This paper (equations 14,15,17) can be seen as one concrete implementation of Hebb's postulate.

---

### Meta-Review · Area_Chair_GeoG · 2022-08-20

**Recommendation:** Accept
**Confidence:** Certain

**Metareview:**

This submission is about brain-inspired learning (Hebbian rules) in the context of kernels. Particularly, given {x^t} points in R^M and a (reproducing) kernel f, the goal of the authors is to learn in a biologically plausible fashion a representation {y^t} in R^N such that f(x^s,x^t) \approx <y^s,y^t> where \approx is meant in squared sense as formulated in (1). They derive the bound (8)-(9) of (1) which is well-suited for Hebbian learning, and apply a stochastic gradient ascent-descent optimization. The practicality of the method is illustrated on the half-moons and the MNIST benchmarks; it performs comparably / favorably to existing approaches and shows sparsity.

Kernel methods are in the forefront of data science. Bringing this field together with online biological updates (Hebbian rules) and extending the neural random Fourier feature (Bahroun et al., 2017) to arbitrary differentiable kernels is a good fit to the focus of NeurIPS with sufficient novelty as it was elaborated by the reviewers.

**Award:**

No

---

### Decision · Program_Chairs · 2022-09-14

Accept